# The Prognostic Role of Body Mass Index on Oncological Outcomes of Upper Tract Urothelial Carcinoma

**DOI:** 10.3390/cancers15225364

**Published:** 2023-11-10

**Authors:** Kang Liu, Hongda Zhao, Chi-Fai Ng, Jeremy Yuen-Chun Teoh, Pilar Laguna, Paolo Gontero, Iliya Saltirov, Jean de la Rosette

**Affiliations:** 1S.H. Ho Urology Centre, Department of Surgery, The Chinese University of Hong Kong, Hong Kong 999077, China; kliu@surgery.cuhk.edu.hk (K.L.); hdzhao@surgery.cuhk.edu.hk (H.Z.); ngcf@surgery.cuhk.edu.hk (C.-F.N.); 2Department of Urology, Medipol Mega University Hospital, Istanbul Medipol University, 34000 Istanbul, Turkey; m.p.lagunapes@gmail.com (P.L.); j.j.delarosette@gmail.com (J.d.l.R.); 3Department of Urology, University of Turin, 10124 Turin, Italy; paolo.gontero@unito.it; 4Department of Urology and Nephrology, Military Medical Academy, 1000 Sofia, Bulgaria; saltirov@vma.bg

**Keywords:** body mass index, upper tract urothelial carcinoma, overall survival, cancer-specific survival, recurrence-free survival

## Abstract

**Simple Summary:**

In patients with upper tract urothelial carcinoma, the impact of body mass index on oncological outcomes is still a matter of debate. We use the Clinical Research Office of the Endourology Society Urothelial Carcinomas of the Upper Tract Registry to compare the overall survival, cancer-specific survival, and recurrence-free survival between normal weight, overweight and obese patients. After balancing the clinicopathological features by propensity score matching, being overweight/obese (body mass index ≥ 25.0 kg/m^2^) was associated with a decreased risk of recurrence in upper tract urothelial carcinoma patients but not overall survival or cancer-specific survival.

**Abstract:**

(1) Objective: The aim of this study was to evaluate whether overweight and obese upper urinary tract carcinoma (UTUC) patients have better or worse survival outcomes. (2) Methods: The Clinical Research Office of the Endourology Society Urothelial Carcinomas of the Upper Tract (CROES-UTUC) Registry was used to extract the data of normal-weight or overweight/obese UTUC patients between 2014 and 2019. Patients with a BMI between 18.5 and 24.9 kg/m^2^ were defined as normal weight, while those with a BMI ≥ 25.0 kg/m^2^ were considered as overweight/obese group. We compared baseline characteristics among groups categorized by different BMIs. The Kaplan–Meier plots with the log-rank test were used to explore the overall survival (OS), cancer-specific survival (CSS), and recurrence-free survival (RFS). Propensity score matching was performed to eliminate the differences in clinicopathologic features. The Declaration of Helsinki was followed during this study. (3) Results: Of 1196 UTUC patients, 486 patients (40.6%) were normal weight, while 710 patients (59.4%) presented with a BMI ≥ 25.0 kg/m^2^. After propensity score matching, all baseline characteristics were balanced. For normal weight and overweight/obese patients, 2-year overall survival rates were 77.8% and 87.2%, 2-year cancer-specific survival rates were 85.2% and 92.7%, and 2-year recurrence rates were 50.6% and 73.0%, respectively. The overweight patients obtained a better RFS (*p* = 0.003, HR 0.548, 95% CI 0.368–0.916) while their OS (*p* = 0.373, HR 0.761, 95% CI 0.416–1.390) and CSS (*p* = 0.272, HR 0.640, 95% CI 0.287–1.427) were similar to normal weight patients. (4) Conclusions: Being overweight/obese (BMI ≥ 25.0 kg/m^2^) was associated with a decreased risk of recurrence in UTUC patients but not overall survival or cancer-specific survival.

## 1. Introduction

Urothelial carcinomas are the sixth most common malignancy in developed countries [1], with upper tract urothelial carcinomas (UTUCs) comprising 5 to 10 percent with an estimated annual incidence of one to two per 100,000 population. There are many differences when comparing UTUCs with its “sibling tumor” of the bladder. Nearly 60% of UTUCs are muscle-invasive, compared with only 15–25% in bladder cancer. Regardless of the tumor’s local staging, the gold standard treatment for localized UTUCs is radical nephroureterectomy [2]. Gender, smoking history, and coronary heart disease were reported to be independent prognostic factors for UTUC patients [3,4]. However, due to the low prevalence, the current prediction model for UTUCs is far from adequate [5]. Hence, identifying relevant preoperative variables aids in the refinement of risk classification and guides proper therapy.

Body mass index (BMI) is a tool used to assess whether a person is overweight or obese, and a high BMI is frequently associated with metabolic issues [6]. Elevated BMI has become a significant public health issue as it has been related to an increased risk of carcinogenesis [7]. While a higher BMI is strongly linked to the development of malignancies such as renal cell carcinoma [8], its role varies in predicting survival outcomes among different cancer types [9,10]. Also, there are inconsistent findings on the impact of elevated BMI on the survival outcomes of UTUC patients [11]. Therefore, we aimed to evaluate the prognostic role of BMI on overall survival (OS), cancer-specific survival (CSS), and recurrence-free survival (RFS) of UTUCs in a contemporary global registry, which can provide further evidence for disease management.

## 2. Method

### 2.1. Study Design and Population

The Clinical Research Office of the Endourology Society Urothelial Carcinomas of the Upper Tract (CROES-UTUC) registry, one of the largest real-world prospective global datasets on the management of UTUCs, was established in 2014. It enrolled patients aged ≥18 years with suspected UTUCs undergoing any diagnostic or surgical intervention among 101 centers from 29 countries [12]. The study criteria were wide-ranging to provide comprehensive real-world data. The registry follows the recommendations of the Agency for Healthcare Research and Quality for the design and use of patient registries for scientific, clinical, and health policy purposes. The study was registered with clinicaltrials.gov (NCT02281188). The Declaration of Helsinki was followed during this study.

The CROES-UTUC registry collected clinical data on baseline characteristics, risk factors, clinical assessment, intervention received, and survival outcomes of the target population. Data from all participating centers were collected using an online Data Management System. The Data Management System was a web-based system located and maintained at the CROES Office.

### 2.2. Statistical Analysis

Data was analyzed using SPSS version 25.0 (SPSS, Inc., Chicago, IL, USA) and R software (4.2.0; R Foundation for Statistical Computing, Vienna, Austria).The primary outcomes of this study were OS, CSS, and RFS. Patients were first dichotomized according to whether their BMI was normal or not. Patients with a BMI between 18.5 and 24.9 kg/m^2^ were defined as normal weight, while those with a BMI ≥ 25.0 kg/m^2^ were considered as overweight/obese group. All data were summarized by descriptive statistics. Frequency distributions of categorical variables are depicted. A chi-square test was employed to compare the differences between groups. Oncological outcomes and BMI-stratified subgroup (overweight: 25.0 kg/m^2^ ≤ BMI < 30.0 kg/m^2^; obese: 30.0 kg/m^2^ ≤ BMI < 35.0 kg/m^2^; severe obese: ≥35.0 kg/m^2^) were analyzed using the Kaplan–Meier plots and the log-rank test. Univariable and multivariate Cox regression analyses were performed, if appropriate. A *p*-value of <0.05 was considered statistically significant. Propensity scores matching (PSM) was performed with a match tolerance of 0.1, the randomization of case order, and priority to nearest matches. Parameters enrolled during the propensity scores matching include variables that had significant differences between normal weight and overweight/obese group (i.e., gender, ethnicity, American Society of Anesthesiologists (ASA) score, diabetes, and hemiplegia) and tumor features (tumor stage, tumor grade, and tumor focality). Some data were missing, and we assumed that the missing pattern was completely random and would not affect our further analysis.

## 3. Results

### 3.1. Patient Demographics

A total of 1196 patients were included in this analysis; of these, 486 patients (40.6%) had normal weight, and the other 710 patients (59.4%) had BMI ≥ 25 kg/m^2^ (Figure 1). Table 1 summarizes the clinicopathologic features of the study population: Notably, 663 (55.4%) patients were aged more than 70,853 (71.3%) were male, 323 (27.0%) were current smokers, 404 (33.8%) were ex-smokers, and 435 (36.4%) were ASA III-V. For the Charlson Comorbidity Index, most of the population (68.1%) received less than five scores. Approximately 70% of the cohort received surgical treatment. Regarding the tumor pathology, 556 (46.5%) patients had pTa/Tis/T1 disease, while 441 (36.9%) patients suffered from muscle-invasive UTUCs, 227 (19.0%) had G1 tumor, and 856 (71.6%) presented with unifocal tumors. Comorbidities and survival outcomes, such as diabetes, renal diseases, all-cause mortality, and recurrence rate, were also listed. 

#### 3.1.1. Comparison of the Baseline Characteristics before Matching

While tumor staging, tumor grade, and tumor focality had no differences between the two groups, the overweight and obese patient group had a high rate of male gender (*p* = 0.020) and white race (*p* < 0.001). Nevertheless, a higher proportion of patients with BMI ≥ 25 kg/m^2^ obtained higher ASA scores (*p* = 0.007) as well as a higher prevalence of diabetes (*p* = 0.049), but a lower prevalence of hemiplegia (*p* = 0.022). Otherwise, all-cause mortality, cancer-specific mortality, and recurrence status did not significantly differ between the groups. 

#### 3.1.2. Comparison of the Baseline Characteristics after Matching

After 1:1 propensity score matching, baseline characteristics were balanced (Appendix A). While gender (Adjust Mean Difference (AMD) = 0.064), ethnicity (AMD < 0.001), ASA scores (AMD = 0.009), diabetes (AMD = 0.022), and hemiplegia (AMD < 0.001) were well matched, eliminating between-group differences (Table 2), the intravesical recurrence rate was significantly higher in the normal weight group (*p* = 0.015). The accumulative recurrence of both the upper tract and bladder depicted the same trend (30.4% in the normal weight group vs. 18.2% in the overweight/obese group, *p* = 0.003) (Appendix A).

#### 3.1.3. Survival Outcomes before Matching

The Kaplan–Meier plots were used to depict OS, CSS, and RFS. Before propensity scores matching, the results indicated that being overweight or obese was associated better OS (*p* = 0.041, HR 0.674, 95% CI 0.460–0.986) and CSS (*p* = 0.048, HR 0.605, 95% CI 0.366–1.000) but not RFS (*p* = 0.156, HR 0.844, 95% CI 0.666–1.068) (Figure 2, Appendix A). According to the Kaplan–Meier plots of subgroup analysis, no significant differences were observed in OS (*p* = 0.328), CSS (*p* = 0.178), and RFS (*p* = 0.829) (Appendix A). The 2-year overall survival rates were 78.4% and 87.4%, the 2-year cancer-specific survival rates were 86.4% and 92.3%, and the 2-year recurrence rates were 56.1% and 65.2% for normal weight and overweight/obese groups, respectively. 

#### 3.1.4. Survival Outcomes after Matching

For normal weight and overweight/obese patients, the 2-year overall survival rates were 77.8% and 87.2%, the 2-year cancer-specific survival rates were 85.2% and 92.7%, and the 2-year recurrence rates were 50.6% and 73.0%, respectively. Overweight/obese patients had a better RFS (*p* = 0.003, HR 0.548, 95% CI 0.368–0.916) than the normal-weight group, while there were no differences in terms of OS (*p* = 0.373, HR 0.761, 95% CI 0.416–1.390) and CSS (*p* = 0.272, HR 0.640, 95% CI 0.287–1.427) (Figure 3). 

## 4. Discussion

Since Ancel Keys et al. introduced “body mass index” (BMI) to describe the ratio of human body weight to squared height in July 1972 [13], BMI has been accepted globally as an indicator for body weight and relative obesity of adults. A BMI of less than 18.5 in adults is considered underweight and may suggest malnutrition, an eating disorder, or other health concerns, whereas a BMI of 25.0 kg/m^2^ or more is considered overweight, and 30.0 kg/m^2^ or more is considered obese [14]. In our study, we divided the research population into two groups: normal weight and overweight/obesity. Our results demonstrated that elevated BMI was associated with a lower risk of recurrence in UTUC patients. Although the results of the original cohort suggest that BMI ≥ 25.0 kg/m^2^ might benefit patients in OS and CSS, our study reveals that BMI has no effect on OS and CSS after propensity score matching.

There are many other studies investigating the role of BMI in UTUC patients. The first published research was conducted by Ehdaie et al. [15]. Notably, 520 cases were included in that study, but there were some unbalanced clinical parameters, such as the ASA score, which was also used in our cohort. After univariate and multivariate Cox regression analysis, normal-weight patients gained the best RFS, CSS, and OS. These contradictory results might be attributed to the fact that patients with elevated BMI are more likely to have lymphovascular invasion and infiltrative architecture. Later, Bachir et al. showed that obesity was associated with worse RFS [16]. Dabi et al. also supported that higher BMI was an independent risk factor for poor CSS and RFS [17]. In our global cohort study, however, a higher BMI (≥25.0 kg/m^2^) appears to be a protective factor for bladder recurrence. To the best of our knowledge, herein we conducted the largest study in this field that provides a real-world perspective to reconsider the real impact of overweight and obesity. It is reasonable that pre-matching results were in line with some previous studies because there were significant differences in the demographical variables and comorbidities. Our propensity score matching revealed robust survival comparison results with higher reliability.

The effects of elevated BMI have been explored and investigated in other urological cancers. Martini et al. found that elevated BMI was a significant predictor of pathologic complete response in patients with muscle-invasive bladder cancer after immunotherapy [18]. Although they did not identify genomic clusters associated explicitly with high BMI and pathologic response at cystectomy, a higher rate of CD8+ lymphocytes was found in the cystectomy specimens from patients with elevated BMI. An elevated BMI has been associated with an increased risk of renal cell carcinoma. At the same time, higher BMI was shown to be a prognostic factor for patients with metastatic clear renal cell carcinoma who were treated with the programmed cell death 1 protein/programmed cell death 1 ligand 1-based immune checkpoint inhibitors [19] or vascular endothelial growth factor targeted therapy [20]. For prostate cancer, a meta-analysis revealed that the hazard ratios for prostate cancer death were 1.07 (95% confidence interval = 0.97–1.17) per 5 kg/m^2^ higher BMI [21]. However, Verma et al. revealed that overweight and obese metastatic castration-resistant prostate cancer patients had a longer OS than lean patients while receiving docetaxel treatment [22]. Still, the elevated BMI paradox exists in prostate cancer patients.

The underlying mechanism of how BMI affects RFS is currently unknown. One hypothesis is that patients with elevated BMI would have more visceral fat surrounding the tumor area, thus avoiding direct invasion to other adjacent structures in the case of locally advanced disease. However, Yeh et al. claimed thicker fat barrier does not help to prevent further invasion of advanced cancer cells or decrease the possibility of residual tumors [6]. BMI and ethnicity could also be closely related, not only in terms of body habitus but also the disease status of UTUCs. The rate of pT3/4 diseases in Asian studies was 37.0–51.3%, whereas in Caucasian studies, it was 27.3–32.1% [14,17,23,24]. In our study, we performed propensity score matching and used a unified standard BMI ≥ 25.0 kg/m^2^ as a categorical cut-off value. In this case, the proportion of Asian patients in both groups was 17.3%, and a similar pT3/4 patients scale was obtained (36% in the regular weight group and 32.9% in the overweight group). Hence, the results are robust and reliable. 

Another potential mechanism is the strong inverse correlation between obesity and decreased serum testosterone [25]. Eriksson et al. used a bidirectional Mendelian randomization analysis to deduce the causal relationship between obesity and serum testosterone status [26]. In short, one unit of standard deviation increase in BMI was associated with a 0.25 standard deviation decrease in serum testosterone. Yeap et al. also confirmed that BMI was inversely related to calculated free testosterone, testosterone, and sex hormone-binding globulin [27]. Traditionally, androgen signaling has been proposed as a possible reason for urothelial carcinoma incidence disparities between men and women. Even without known risk factors, men have a threefold increased risk of urothelial carcinoma compared to women [28]. This indicates that testosterone has a significant effect in promoting urothelial cancer. Thus, an elevated BMI combined with a low testosterone level and less recurrence could explain our findings. 

One of the strengths of this study is that it is the largest UTUC cohort, which allows effect sizes to be assessed with better precision. Propensity score matching was performed to balance the clinicopathological parameters and reveal reliable results. The first limitation of this study is its observational nature, which is an inherent limitation. Secondly, there can be missing data in our cohort, which could be compensated for by the considerable sample size. Thirdly, it is a heterogeneous cohort due to real-world data collection that we adjusted through propensity score matching. 

## 5. Conclusions

BMI ≥ 25.0 kg/m^2^ was associated with a decreased risk of recurrence in UTUC patients but not overall survival or cancer-specific survival. This new information may be useful to tailor the surveillance protocol in a more individualized manner. 

## Figures and Tables

**Figure 1 cancers-15-05364-f001:**
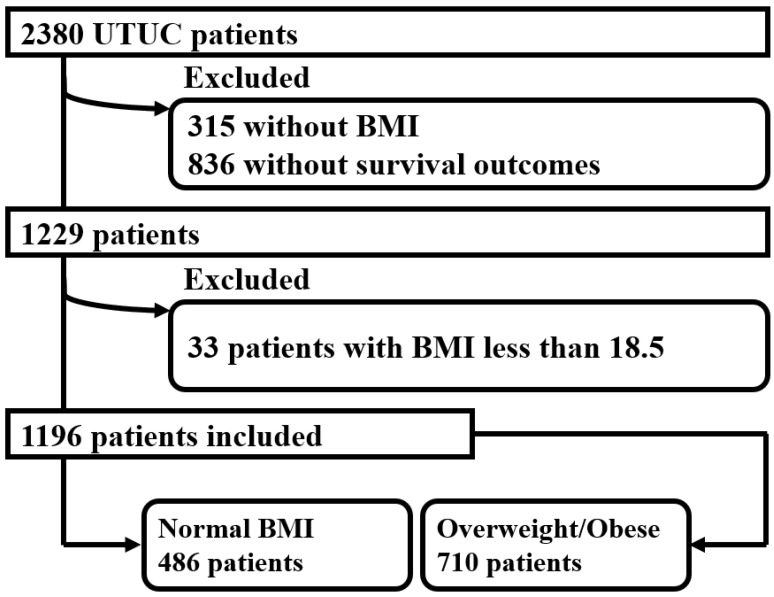
Flowchart for development of the dataset.

**Figure 2 cancers-15-05364-f002:**
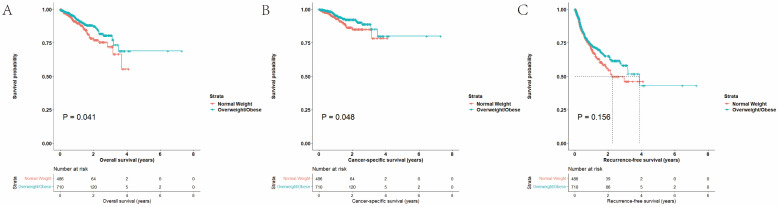
Survival outcomes of the whole cohort: (**A**) OS, (**B**) CSS, and (**C**) RFS.

**Figure 3 cancers-15-05364-f003:**
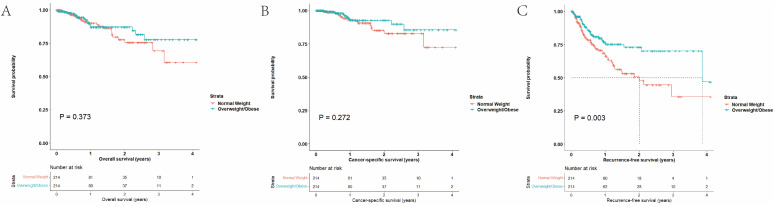
Survival outcomes after PSM: (**A**) OS, (**B**) CSS, and (**C**) RFS.

**Table 1 cancers-15-05364-t001:** Baseline characteristics between the two BMI groups.

Characteristics	Normal Weight *n* = 486 *n* (%)	Overweight/Obese *n* = 710 *n* (%)	Total *n* = 1196 *n* (%)	*p*-Value
Age				
<70	210 (43.2)	319 (44.9)	529 (44.3)	0.596
≥70	274 (56.4)	389 (54.8)	663 (55.4)	
Missing	2 (0.4)	2 (0.3)	4 (0.3)	
Gender				
Female	157 (32.3)	185 (26.1)	342 (28.6)	0.020
Male	329 (67.7)	524 (73.8)	853 (71.3)	
Missing	0 (0.0)	1 (0.1)	1 (0.1)	
Smoking status				
No	152 (31.3)	219 (30.8)	371 (31.0)	0.790
Ex-smoker	156 (32.1)	248 (34.9)	404 (33.8)	
Current smoker	130 (26.7)	193 (27.2)	323 (27.0)	
Missing	48 (9.9)	50 (7.1)	98 (8.2)	
Ethnicity				
White	343 (70.6)	594 (83.7)	937 (78.3)	<0.001
Asian	115 (23.6)	55 (7.7)	170 (14.2)	
Other	17 (3.5)	45 (6.3)	62 (5.2)	
Missing	11 (2.3)	16 (2.3)	27 (2.3)	
Previous Malignancies				
No	325 (66.9)	443 (62.4)	768 (64.2)	0.075
Yes	145 (29.8)	248 (34.9)	393 (32.9)	
Missing	16 (3.3)	19 (2.7)	35 (2.9)	
Occupational Hazard				
No	392 (80.7)	547 (77.0)	939 (78.5)	0.163
Yes	8 (1.6)	20 (2.8)	28 (2.3)	
Missing	86 (17.7)	143 (20.2)	229 (19.2)	
Family history				
No	349 (71.8)	492 (69.3)	841 (70.4)	0.246
Yes	29 (6.0)	54 (7.6)	83 (6.9)	
Missing	108 (22.2)	164 (23.1)	272 (22.7)	
ASA				
I–II	320 (65.8)	412 (58.0)	732 (61.2)	0.007
III–V	155 (31.9)	280 (39.4)	435 (36.4)	
Missing	11 (2.3)	18 (2.5)	29 (2.4)	
Charlson Comorbidity Index				
0	104 (21.4)	161 (22.7)	265 (22.2)	0.628
1–2	137 (28.2)	237 (33.4)	374 (31.3)	
3–4	73 (15.0)	103 (14.5)	176 (14.6)	
5–10	23 (4.7)	44 (6.2)	67 (5.6)	
Missing	149 (30.7)	165 (23.2)	314 (26.3)	
Procedure				
RNU	323 (66.5)	450 (63.4)	773 (64.6)	0.480
KSS	27 (5.5)	45 (6.3)	72 (6.0)	
Missing	136 (28.0)	215 (30.3)	351 (29.4)	
Surgical Margin				
Negative	309 (63.6)	433 (61.0)	742 (62.0)	0.839
Positive	25 (5.1)	37 (5.2)	62 (5.2)	
Missing	152 (31.3)	240 (33.8)	392 (32.8)	
Tumour Size				
<1 cm	30 (6.1)	57 (8.0)	87 (7.3)	0.102
1–2 cm	114 (23.5)	178 (25.1)	292 (24.4)	
>2 cm	65 (13.4)	71 (10.0)	136 (11.4)	
Missing	277 (57.0)	404 (56.9)	681 (56.9)	
TNM (2009) Staging—Left and Right				
pTa	133 (27.4)	209 (29.4)	342 (28.6)	0.647
pTis	6 (1.2)	12 (1.7)	18 (1.5)	
pT1	83 (17.1)	113 (15.9)	196 (16.4)	
pT2	76 (15.6)	89 (12.5)	165 (13.8)	
pT3	107 (22.0)	146 (20.6)	253 (21.2)	
pT4	11 (2.3)	12 (1.7)	23 (1.9)	
Missing	70 (14.4)	129 (18.2)	199 (16.6)	
Tumor Grade				
G1	89 (18.3)	138 (19.5)	227 (19.0)	0.653
G2	105 (21.6)	139 (19.5)	244 (20.4)	
G3	209 (43.0)	284 (40.0)	493 (41.2)	
Missing	83 (17.1)	149 (21.0)	232 (19.4)	
Multifocal Tumor				
No	339 (69.7)	517 (72.8)	856 (71.6)	0.636
Yes	80 (16.5)	113 (15.9)	193 (16.1)	
Missing	67 (13.8)	80 (11.3)	147 (12.3)	
Comorbidities				
Diabetes	68 (14.0)	140 (19.7)	208 (17.4)	0.049
Renal disease	58 (11.9)	108 (15.2)	166 (13.9)	0.294
Liver disease	13 (2.7)	10 (1.4)	23 (1.9)	0.073
Congestive heart failure	23 (4.7)	33 (4.6)	56 (4.7)	0.687
Myocardial infarction	45 (9.3)	73 (10.3)	118 (9.9)	0.925
Chronic pulmonary disease	53 (10.9)	109 (15.4)	162 (13.5)	0.094
Cerebrovascular disease	29 (6.0)	38 (5.4)	67 (5.6)	0.402
Peripheral vascular disease	44 (9.1)	64 (9.0)	108 (9.0)	0.615
Ulcer disease	7 (1.4)	19 (2.7)	26 (2.2)	0.217
Dementia	15 (3.1)	24 (3.4)	39 (3.3)	0.993
Hemiplegia	4 (0.8)	0 (0.0)	4 (0.3)	0.022
Connective tissue disease	3 (0.6)	11 (1.5)	14 (1.2)	0.185
AIDS	2 (0.4)	1 (0.1)	3 (0.3)	0.562
All-Cause Mortality	51 (5.9)	55 (4.8)	106 (5.3)	0.101
Cancer-specific Mortality	31 (3.6)	30 (2.6)	61 (3.0)	0.096
Recurrence-UTUC and Bladder	121 (14.0)	162 (22.8)	283 (23.7)	0.406
Recurrence of UTUC	53 (6.1)	71 (10.0)	124 (10.4)	0.614
Recurrence of Bladder	85 (9.9)	117 (10.1)	202 (10.0)	0.647

ASA: American Society of Anaesthesiologists; RNU: Radical nephroureterectomy; KSS: Kidney-sparing surgery.

**Table 2 cancers-15-05364-t002:** Matched characteristics between the two BMI groups.

Characteristics	Normal Weight *n* = 214 *n* (%)	Overweight/Obese *n* = 214 *n* (%)	Total *n* = 428 *n* (%)	AMD
Gender				
Female	61 (28.5)	66 (30.8)	127 (29.7)	0.064
Male	153 (71.5)	148 (69.2)	301 (70.3)	
Ethnicity				
White	177 (82.7)	177 (82.7)	177 (41.4)	<0.001
Asian	30 (14.0)	30 (14.0)	30 (7.0)	
Other	7 (3.3)	7 (3.3)	7 (1.6)	
ASA				
I–II	124 (57.9)	120 (56.1)	244 (57.0)	0.009
III–V	90 (42.1)	94 (43.9)	184 (43.0)	
TNM (2009) Staging—Left and Right				
pTa	85 (39.7)	88 (41.1)	173 (40.4)	0.077
pTis	1 (0.5)	2 (0.9)	3 (0.7)	0.042
pT1	40 (18.7)	40 (18.7)	80 (18.7)	<0.001
pT2	34 (15.9)	38 (17.8)	72 (16.8)	<0.001
pT3	52 (24.3)	45 (21.0)	97 (22.7)	0.089
pT4	2 (0.9)	1 (0.5)	3 (0.7)	0.036
Tumor Grade				
G1	51 (23.8)	57 (26.6)	108 (25.2)	0.077
G2	59 (27.6)	63 (29.4)	122 (28.5)	0.063
G3	104 (48.6)	94 (49.3)	198 (46.3)	0.009
Multifocal Tumor				
No	169 (79.0)	167 (78.0)	336 (78.5)	0.046
Yes	45 (21.0)	47 (22.0)	92 (21.5)	
Comorbidities				
Diabetes	36 (16.8)	39 (18.2)	75 (17.5)	0.022
Hemiplegia	0 (0.0)	0 (0.0)	0 (0.0)	<0.001

ASA: American Society of Anaesthesiologists; AMD: Adjust Mean Difference.

## Data Availability

The datasets used and/or analyzed during the current study are available from the corresponding author upon reasonable request.

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
