# Peer review of "The Prognostic Role of Body Mass Index on Oncological Outcomes of Upper Tract Urothelial Carcinoma"

_cancers, 2023, doi:10.3390/cancers15225364_

Round 1
Reviewer 1 Report
Comments and Suggestions for Authors
In this study, authors have compared survival outcomes by BMI in patients with UTUC.
They found that patients with higher BMI presented a lower rate of recurrence.
I have some points that require further clarification.
In general
-BMI is not a direct measure of obesity and other methodologies such as bioimpedance would be more appropriate such as CT scan or bioimpedance.
-It is not the same a patient with BMI 25 than other with BMI 40, a stratified analysis would be required ( I do not know if possible due to sample size)
Abstract
chrematistics? do you mean characteristics?
A more detailed information on the dataset would be helpful for understanding the sample. For example
In table 1, age, smoking status do not sum 862 example (277+250+237) The same in overweight group and total in many categories. Maybe the authors are not including not available data. Please check.
The same in table 2 with occupational hazard, family history, tumor size, and others. Maybe the authors are not including not available data. Please check.
What is the median follow-up?
Was follow-up standarized among centers? How was it performed?
Kaplan Meier curves should have number at risk patients.
Please add the HR values?
How RFS was calculated? local recurrence at the kidney fossa? In the remaining part in case of KSS? bladder recurrence?
In limitation paragraph state
Second, there can be missing data in our cohort that could be compensated for by the considerable sample size.
-In which variables there is missing data?
Comments on the Quality of English LanguageAbstract
chrematistics? do you mean characteristics?
Reviewer 2 Report
Comments and Suggestions for Authors
Abstract; Background „Whether overweight and obese upper urinary tract carcinoma (UTUC) patients have better or worse survival outcomes remain controversial.” Please write the full sentence, for example: “Aim of this study was to evaluate whather…..”
Statistical analyses: what is with patients with BMI <18.5; were they excluded?
Is detailed data on BMI available? I ≥25.0 is too bride term; this should be classified in overweight (25-29), obese (30-34.9), severe obese (>=35), for example.
Figure are very small; readers must strongly increase the page size to see them. Please modify.
Authors should conduct Cox regression analyses (even univariable, not necessary multivariable as cohorts were well matched) to show the Hazard Ratios for BMI. This would strengthen the study results and conclusions.
Round 2
Reviewer 2 Report
Comments and Suggestions for Authors
Thank you for responses